# LEARNING MOVEMENT STRATEGIES FOR MOVING TARGET DEFENSE

## ABSTRACT

The field of cybersecurity has mostly been a cat-and-mouse game with the discovery of new attacks leading the way. To take away an attacker's advantage of reconnaissance, researchers have proposed proactive defense methods such as Moving Target Defense (MTD). To find good movement strategies, researchers have modeled MTD as leader-follower games between the defender and a cyber-adversary. We argue that existing models are inadequate in sequential settings when there is incomplete information about a rational adversary and yield sub-optimal movement strategies. Further, while there exists an array of work on learning defense policies in sequential settings for cyber-security, they are either unpopular due to scalability issues arising out of incomplete information or tend to ignore the strategic nature of the adversary simplifying the scenario to use single-agent reinforcement learning techniques. To address these concerns, we propose (1) a unifying game-theoretic model, called the Bayesian Stackelberg Markov Games (BSMGs), that can model uncertainty over attacker types and the nuances of an MTD system and (2) a Bayesian Strong Stackelberg Q-learning (BSS-Q) approach that can, via interaction, learn the optimal movement policy for BSMGs within a reasonable time. We situate BSMGs in the landscape of incomplete-information Markov games and characterize the notion of Strong Stackelberg Equilibrium (SSE) in them. We show that our learning approach converges to an SSE of a BSMG and then highlight that the learned movement policy (1) improves the state-of-the-art in MTD for web-application security and (2) converges to an optimal policy in MTD domains with incomplete information about adversaries even when prior information about rewards and transitions is absent.

## 1 INTRODUCTION

The complexity of modern-day software technology has made the goal of deploying fully secure cyber-systems impossible. Furthermore, an attacker often has ample time to explore a deployed system before exploiting it. To level the playing field, researchers have introduced the idea of proactive cyber defenses such as Moving Target Defense. In Moving Target Defense (MTD), the defender shifts between various configurations of the cyber-system (1). This makes the attacker's knowledge, gathered during the reconnaissance phase, useless at attack time as the system may have shifted to a new configuration in the window between reconnaissance and attack. To ensure that an MTD system is effective at maximizing security and minimizing the impact on the system's performance, the consideration of an optimal movement strategy is important (2; 3).

MTD systems render themselves naturally to a game-theoretic formulation– modeling the cyber-system as a two-player game between the defender and an attacker is commonplace. The expectation is that the equilibrium of these games yields an optimal (mixed) strategy that guides the defender on how to move their dynamic cyber-system in the presence of a strategic and rational adversary. The notion of Strong Stackelberg Equilibrium predominantly underlies the definition of optimal strategies in these settings (4; 5) as the defender deploys a system first (acting as a leader) while the attacker, who seeks to attack the deployed system, assumes the role of the follower. In many real-world scenarios, single-stage normal-form games do not provide sufficient expressiveness to capture the switching costs of actions (4; 6) or reason about the adversary's sequential behavior (7; 8). On the other hand, works that consider modeling the MTD as a multi-stage stochastic game (9; 10; 11; 8), do not model incomplete information about adversaries, a key aspect of the single-stage normal-form

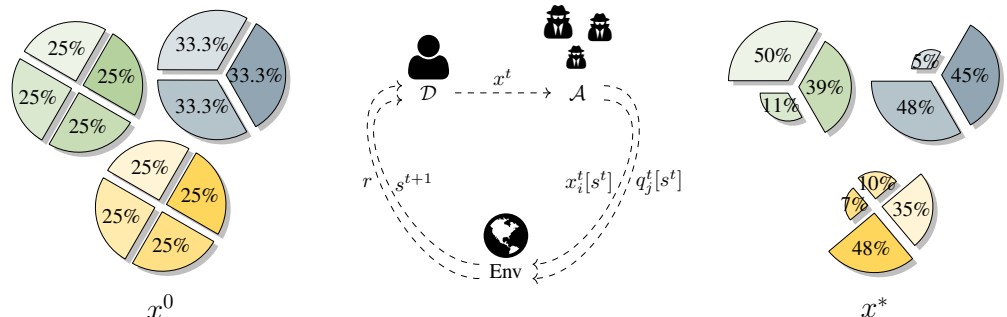

Figure 1: The defender starts with an uniform random strategy ($x^0$); it switches to a possible configuration of a software system with equal probability in each state. Then, the defender, upon interactions with an environment and simulation of an adversary in their head, adapts its strategy at every step and finally converges to the Strong Stackelberg Eq. (SSE) of the BSMG yielding $x^*$.

formalism (known as Bayesian Stackelberg Games (BSG) (12; 6)). To address these concerns about expressiveness, while remaining scalable for use in cyber-security settings, we propose the unifying framework of Bayesian Stackelberg Markov Games (BSMG). We show that BSMGs can be used to model various Moving Target Defense scenarios, capturing the uncertainty over attacker types and sequential impacts of attacks and switching defenses. We characterize the notion of optimal strategy as the Strong Stackelberg Equilibrium of BSMGs and show that the robust (movement) strategy improves the state-of-the-art found by previous game-theoretic modeling.

While multi-stage game models are ubiquitous in security settings, expecting experts to provide detailed models about rewards and system transitions is considered unrealistic. Thus, researchers have considered techniques in reinforcement learning to learn optimal movement policies over time (13; 14; 15; 16). Unfortunately, these works ignore (1) the strategic nature and the rational behavior of an adversary and (2) the incomplete knowledge a defender may possess about their opponent. This, as we show in our experiments, results in a new attack surface where the defender's movement policy can be exploited by an adversary. To mitigate this, we bridge the knowledge gap between existing work, and techniques in multi-agent reinforcement learning by proposing a Bayesian Strong Stackelberg Q-learning (BSS-Q) approach (graphically shown in Figure 1). First, we can show that BSS-Q converges to the Strong Stackelberg Equilibrium of BSMGs. Second, we design an Open-AI gym (17) style multi-agent environment for two Moving Target Defenses (one for web-application and the other for cloud-network security) and compare the effectiveness of policies learned by BSS-Q against existing state-of-the-art static policies and other reinforcement learning agents.

In the next section, we motivate the need for a unifying framework and formally describe the proposed game-theoretic model of BSMGs. We briefly discuss how two Moving Target Defenses are modeled as BSGMs. We then introduce the Bayesian Strong Stackelberg Q-learning approach and show that it converges to the SSE of BSMGs, followed by a section showcasing experimental results. Finally, before concluding, we discuss related work.

## 2  BAYESIAN STACKELBERG MARKOV GAMES (BSMGS)

Markov Games (MGs) (18) are used to model multi-agent interactions in sequential planning problems. Under this framework, a player can reason about the behavior of other agents (co-operative or adversarial) and come up with policies that adhere to some notion of equilibrium (where no agent can gain by deviating away from the action or strategy profile). While MGs have been widely used to model adversarial scenarios, they suffer from two major shortcomings– (1) they do not consider incomplete information about the adversary (19; 7; 20; 13) and/or (2) they consider weak threat models where the attacker has no information about the defender's policy (21; 14). On the other hand, Bayesian Stackelberg Games (22; 6) are a single-stage game-theoretic formalism that addresses both of these concerns but cannot be trivially generalized to sequential settings.

To overcome these challenges of expressiveness required for Moving Target Defenses (MTDs) while ensuring scalability, we introduce the formalism of Bayesian Stackelberg Markov Games (BSMGs). BSMGs extend Bayesian Stackelberg Games (BSGs) to multi-stage sequential games. While one can consider using existing formalism in Markov Games that capture incomplete information, they face severe scalability issues and have thus been unpopular in cyber-security domains (we discuss how BSMG is situated in this landscape of models in section 5). In the context of Moving Target Defense (MTD), BSMG acts as a unifying framework helping us characterize optimal movement policies against strategic adversaries, capture transition dynamics and costs of the underlying cyber-system, aid in reasoning about stronger threat models, and consider incomplete information about strategic adversaries. Formally, a BSMG can be represented by the tuple $(P, S, \Theta, A, \tau, U, \gamma^{\mathcal{D}}, \gamma^{\mathcal{A}})$ where,

- $P = \{\mathcal{D}, \mathcal{A} = \{\mathcal{A}_1, \mathcal{A}_2, \dots \mathcal{A}_t\}\}$ where $\mathcal{D}$ denotes the leader (defender) and $\mathcal{A}$ denotes the follower (attacker). In our model, only the second player has $t$ types.

- $S = \{s_1, s_2, \dots, s_k\}$ are $k$ (finite) states of the game,

- $\Theta = \{\theta_1, \theta_2, \dots \theta_k\}$ denotes $k$ probability distributions (for $k$ states) over the $t$ attackers and $\theta_i(s)$ denotes the probability of $i$-th attacker type in state $s$

- $A = \{A^{\mathcal{D}}, A^{\mathcal{A}_1}, \dots A^{\mathcal{A}_t}\}$ denotes the action set of the player and $A^i(s)$ represents the set of actions/pure strategies available to player $i$ in state $s$.

- $\tau^i(s, a^{\mathcal{D}}, a^{\mathcal{A}_i}, s')$ represents the probability of reaching a state $s' \in S$ from the state $s \in S$ when the $\mathcal{D}$ chooses $a^{\mathcal{D}}$ and attacker type $i$ choose the action $a^{\mathcal{A}_i}$,

- $U = \{U^{\mathcal{D}}, U^{\mathcal{A}_1}, \dots, U^{\mathcal{A}_t}\}$ where $U^{\mathcal{D}}(s, a^{\mathcal{D}}, a^{\mathcal{A}_i})$ and $U^i(s, a^{\mathcal{D}}, a^{\mathcal{A}_i})$ represents the reward/utility of $\mathcal{D}$ and an attacker type $\mathcal{A}_i$ respectively if, in state $s$, actions $a^{\mathcal{D}}$ and $a^{\mathcal{A}_i}$ are chosen by the players,

- $\gamma^i \mapsto [0, 1)$ is the discount factor for player $i$. We will assume that $\gamma^{\mathcal{D}} = \gamma^{\mathcal{A}_\rangle} = \gamma$.

In BSMGs the individual stage games constitute normal-form Bayesian games with a distribution over attacker types; this is in contrast to the unit probability over a single adversary type in MGs. Both in physical-security (22) and cyber-security (6) domains, defenders are known to have knowledge about follower types, a classic case of *known-unknowns*. BSMGs provide the expressive power to represent this information; precisely $\theta_s$ represents the probability estimate with which a defender believes a certain kind of adversary is encountered in a particular state $s$ of the game.

Note that a defender $\mathcal{D}$ is expected to deploy a system first. Thus, a strong threat model assumes that all the attacker types $\mathcal{A}_i$ know the defender's policy, making the Bayesian notion of Stackelberg Equilibrium an appropriate solution concept for such games. For a normal-form game, let a defender's mixed policy be denoted as $x$ and let us denote an attacker type $\mathcal{A}_i$'s response set (i.e. a set of best responses to $x$) as $R^i(x)$. If the response set for all adversary types is singleton, then the action profile $(x, R^1(x), \dots R^t(x))$ constitutes a Stackelberg Equilibrium of the normal-form game (23). When the response set contains more than one action, the final response chosen can yield different rewards for $\mathcal{D}$. In such cases, a popular assumption made in general-sum games is to consider the response that results in the optimal rewards for $\mathcal{D}$; this is termed as the Strong Stackelberg Equilibrium (SSE) (24; 22; 4; 6). In contrast to the notion of Weak Stackelberg Equilibrium, which considers the pessimistic case, an SSE is guaranteed to exist and yields a unique game value to the defender regardless of the particular SSE chosen (25; 26). Thus, we consider SSEs as the solution concept in BSMGs and highlight a few properties about player strategies at equilibrium for BSMGs (the proofs are deferred to Appendix A).

**Lemma 1.** For a given policy of the leader/defender in BSMG, every follower/attacker type will have a deterministic policy in all states $s \in S$ that is an optimal response.

**Corollary 1.** For an SSE policy of the defender, denoted as $x$, each attacker type $A_i$ has a deterministic best policy $q_i$. The action profile $(x, q_1, \dots q_t)$ denotes the SSE of the BSMG.

**Lemma 2.** If an action profile $(x, q_1, \dots, q_t)$ yields the equilibrium values $V_{x,q}^{\mathcal{D}}$ and $V_{x,q}^{\mathcal{A}_i}$ to the players and is an SSE of BSMG, then $\forall s \in S$ $(x(s), q_1(s), \dots, q_t(s))$ is an SSE of the bi-matrix Bayesian game represented by the Q-values $Q_{x,q_i}^{\mathcal{D},i}(s), Q_{x,q_i}^{\mathcal{A}_i}(s)$ $\forall i \in \{1, \dots, t\}$.

When the parameters of a game are provided up-front, an approach similar to calculating Strong Stackelberg Equilibrium in Bayesian Games (12) alongside Mixed-Integer Non-Linear Programming

approaches (7) or Bellman-style approaches for Markov Games (20) can be leveraged to find the defender's policy. In contrast, when game-parameters are difficult to provide upfront but interaction with an environment is considered possible, we can resort to reinforcement learning techniques. Before proposing our model-free multi-agent reinforcement learning method in the next section, we briefly discuss how the various MTDs, used later in the experiments, are modeled as BSMGs.

## 2.1 Modeling Moving Target Defense Scenarios as BSMGs

A Moving Target Defense (MTD) is defined by the tuple $\langle C, T, M \rangle$ where $C$ represents the set of configurations a system can choose to be in, $T$ represents a timing function that determines when a system switches and $M$ represents a movement function that determines the movement policy (3). We assume a constant function $T$ for switching (the game clock) and discuss how we can leverage $C$ to model the states and the actions of our BSMG. Finally, we leverage existing knowledge, discussed in literature, to model the follower types for a particular MTD scenario.

### 2.1.1 MTD for web-application security

In (6), the authors model an MTD for web-applications as a Bayesian Stackelberg Game (BSG) (22). In addition, they consider the performance impact of movement between configurations (downtime, service latency, etc.) when coming up with a movement policy. Given this utility is a function of a two-stage strategy, this information can't be accurately modeled by a single-stage normal-form BSG that results in a state-agnostic sub-optimal policy. Clearly, the BSMG, given its sequential nature, is better suited to capture this information.

The BSMG has $|C|$ states, each representing a configuration of the MTD system. Each configuration has an equal probability of being the start state in an episode and there exists no terminal state. In each state, $s \in S, A^{\mathcal{D}}(s) = C$, i.e. the defender can choose to move to any configuration (this includes remaining in the same state). The three attacker types are denoted as $\mathcal{A} = \{\mathcal{A}_1, \mathcal{A}_2, \mathcal{A}_3\}$ and the probability distribution, provided in (6), over these types in-state $s$ represented is $\theta_s$. The adversary's action sets, denoted by $(A^{\mathcal{A}_1}, A^{\mathcal{A}_2}, A^{\mathcal{A}_3})$, represent mined CVEs from the National Vulnerability Database (27) (similar to (6)). As per the original domain, the distribution over attacker types (and the follower's attack set) remain the same in all states of the BSMG.

### 2.1.2 MTD against multi-stage attacks

MTD for cloud networks has modeled the problem of placing Intrusion Detection Systems (IDSs) as a Markov Game (21; 20). The key objective of these works is to minimize the performance impact of a deployed defense configuration while ensuring security is maximized. While BSMGs allow us to represent uncertainty over attacker types, existing formulations consider single follower types, thereby boiling down to an MG. Although we will use an MG for this scenario, we note that our framework is capable of incorporating ongoing research on the characterizing attacker types in the context of cyber-systems (28).

Attack graphs are an attack representation method used to capture possible attack paths through a network (29). Nodes of this attack graph, that describe physical locations in the cloud system and an attacker's privilege level, constitute the states of our BSMG, similar to the MG formulation in (20). The defender's actions in a state $A^{\mathcal{D}}(s)$ represents IDS that can be deployed in the state $s$ to identify possible exploits and the attacker's actions represent the possible exploits, which are obtained by considering CVEs effective against the defender's cloud system.

## 3 Bayesian Strong Stackelberg Q-learning in BSMGs

While game-theoretic formalism has been used to model various cyber-security scenarios (30; 31; 6), it is impractical to expect security experts to provide the parameters of the game upfront (13; 15; 16). In the context of Moving Target Defense (MTD) in particular, determining the impact of various attacks, the asymmetric impacts of a particular defense on performance, and the switching cost of a system are better obtained via interaction with an environment. Further, there exists uncertainty regarding the success of an attack (eg. a buffer overflow attack may need significant tinkering for it to

---

**Algorithm 1:** Bayesian Strong Stackelberg Q-learning (BSS-Q) for BSMGs.

---

1: *In:* $(P, S, A, \Theta, \gamma)$, `mtd_sim`, $\alpha$, num_episodes
2: *Out:* Policies of the players $x$ for $\mathcal{D}$, $q^i \; \forall i \in \mathcal{A}$
3: **while** num_episodes $> 0$ **do**
4:     $s \leftarrow$ sample start state from $S$
5:     **while** $s \neq$ terminal state OR !max_eps_len **do**
6:         $i \leftarrow$ sample attacker type using $\theta_s$ from $A$
7:         $a^{\mathcal{D}}, a^{\mathcal{A}_i} \leftarrow \epsilon$-greedy sampling form $x(s)$, $q_i(s)$
8:         $r^{\mathcal{D}}, r^{\mathcal{A}_i}, s' \leftarrow$ `mtd_sim`.act$(s, a^D, a^A)$
9:         $Q^{\mathcal{D},i} \leftarrow (1-\alpha)Q^{\mathcal{D},i}(s, a^{\mathcal{D}}, a^{\mathcal{A}_i}) + \alpha[r^{\mathcal{D}} + \gamma^{\mathcal{D}} V^{\mathcal{D}}(s')]$
10:        $Q^{\mathcal{A}_i} \leftarrow (1-\alpha)Q^{\mathcal{A}_i}(s, a^{\mathcal{D}}, a^{\mathcal{A}_i}) + \alpha[r^{\mathcal{A}_i} + \gamma^{\mathcal{D}} V^{\mathcal{A}_i}(s')]$
11:        $(x, q_j), (V^{\mathcal{D}}, V^{\mathcal{A}_j}) \leftarrow$ solve Bayesian Stackelberg Game$(Q^{\mathcal{D},i}, Q^{\mathcal{A}_j}) \forall j$
12:     **end while**
13: **end while**

---

be successful) and also the success of defense mechanisms (eg. Intrusion Detection Systems based on machine learning can be inaccurate) which can be better inferred via repeated interactions.

In existing works on MTD, the goal, in the presence of (1) game parameters and (2) incomplete information about an adversary, is to learn a robust policy that works best, in expectation, against all adversaries. When modeled as a multi-agent reinforcement learning problem, an interesting distinction ensues. If the defender gets to interact with the environment and an actual adversary, they can update their incomplete information about the adversary, leading to a Bayesian style update regarding the attacker types in the new state returned by the environment (14). Unfortunately, reinforcement learning methods, which are often sample-inefficient, will require abundant interaction with a real-world adversary; an impossible feat in cyber-security scenarios. Thus, the best the defender can do is to simulate a rational adversary in their head. Given the adversary's type is unknown during the policy learning phase, the defender needs to sample an attacker from the set of attacker types in each state of the game to eventually learn a robust policy in each state.

To learn a robust policy, we consider the use of a Multi-agent Reinforcement Learning approach for BSMG. Specifically, given the inherent leader-follower paradigm present in our setting, we propose the use of a Bayesian Strong Stackelberg Q-learning (BSS-Q) approach for BSMGs discussed in algorithm 1. The approach is similar to existing work in multi-agent reinforcement learning (MARL) and considers a Bellman-style Q-learning approach for calculating the agent policies over time. In lines 9 and 10, we update the Q-values for the players $\mathcal{D}$ and adversary type $\mathcal{A}_i$ in the state $s$ using the rewards obtained by acting in the environment. Since we simulate the adversary, we can select an action on its behalf and send it across to the simulator `mtd_sim`. Given `mtd_sim` which has an idea whether the attack succeeded or failed, it can send us back the attacker's reward.[1] In existing works on MARL, they make a default assumption that the defender gets the attacker's reward even when the adversary is not simulated (32). While this assumption is somewhat justified in the context of constant-sum or common-payoff games, it becomes unrealistic in the context of general-sum games.

In line 11, we use a Bayesian Stackelberg Game solver to calculate the BSS of the normal form Bayesian bi-matrix game defined by the Q-values in state $s$. It is known that finding an SSE of a Bayesian Stackelberg Game is NP-hard and thus, the computation in line 10 might seem prohibitive. In practice, as shown in our experiments, compact representation of the scenario as a MILP (22) can help in computing the value and the policy within a second even for web-application domains with more than 300 executable attack actions (6). Note that even though only one follower type acts in the environment, the change in the defender's policy because of this can lead to the other follower types switching their actions. Hence, solving the Bayesian stage game (in its full glory) becomes essential to converge to an SSE policy of the BSMG (and batch update methods, which can speed-up computation, become less reliable). Methods that scale-up equilibrium computation in security games (4) rely on a known reward structure. Unfortunately, this makes it difficult to justify their use in our setting, where the rewards are unknown and thus, may have an arbitrary reward structure.

---

[1]If interaction with an adversary is possible, the defender should consider a Bayesian style update of the parameters $\theta_s$ depending on the observed action and the observed reward after line 10.

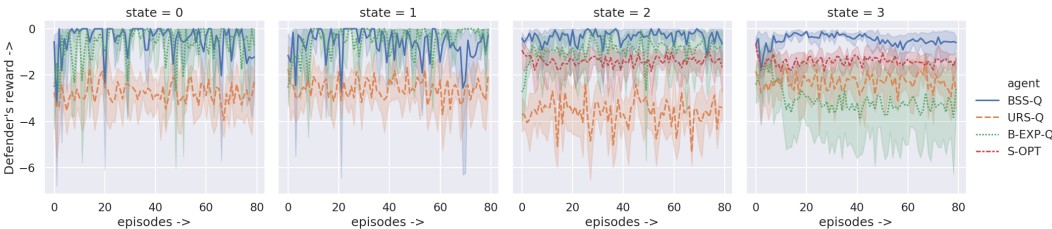

Figure 2: The defender's value in the four states of the BSMG modeling the MTD for web-applications when using BSS Q-learning (blue) compared to Uniform Random Strategy (orange) (34), a Bayesian version of EXP-Q learning (green) (13) and the optimal stage-agnostic strategy (red) (6).

Note that one can choose to move the attacker type sampling (Line 6) outside the inner while loop. This would imply sampling an attacker for each episode and interacting with them, making it easier to understand that the attacker type can plan for future rewards (when $\gamma^{A_i} > 0$) and come up with far-sited strategies. Given that an attacker type used it own Q-value and value functions for updating its Q-values (Line 10), sampling done is each state (i.e. inside the nested while loop) still ensures that the attacker types are not myopic. Further, sampling in each states results in the defender not over-fitting to one attacker in one episode and then having to adjust it policy for a different attacker in the next one which can lead to a slower convergence. Now, we show that our method converges to the optimal policy at Bayesian Strong Stackelberg Equilibrium.

**Proposition 1. (Convergence Result)** *The Bayesian Strong Stackelberg Q-learning approach converges to the BSS of a BSMG.* (The detailed proof can be found in Appendix A.)

*Proof Sketch.* We show that for all the players (i.e. the defender and the individual attacker types) (1) the Q-value calculation (that uses the game values of the future states, denoted as an operator) is a *real contraction operator* and the (2) update process (Line 9-10 of algorithm 1) converges to a fixed point. While the proof skeleton is similar to the one in (33), the proof is more complex due to the presence of multiple attacker types. In proving (1), we need to reason about Bayesian stage games. □

## 4 EXPERIMENTS

We conduct experiments to understand the effectiveness of the learned movement policies for two MTD scenarios. As many existing baselines can't handle unknown utilities and transition dynamics, we develop an OpenAI style (17) game simulator that is aware of the underlying game parameters but interacts with the learning agents only via selected public-facing APIs. This helps us to compare against baselines that assume game parameters are available. While the impacts of attack actions are obtained in the simulator using the Common Vulnerability Scoring Service (CVSS), we can consider real system interaction, pending investigation, to obtain less-informative and sparse rewards. More details about the game simulator and additional experiments can be found in the supplementary material. For both experiments, the defender (who samples a follower type in each interaction) is a single-thread process and regardless of their own policy, are pitted against a strategic and ration adversary. The code used Gurobi for solving the Bayesian Stackelberg Game (BSG) game in line 11 of algorithm 1 and ran on an Intel Xeon(R) CPU E5-2643 v3 @ 3.40GHz with 64GB RAM.

MTD FOR WEB-APPLICATIONS

We use the game parameters obtained in (6) to design our game simulator. The simulator, given the current state and the defender's action, lets us return the switching costs as a part of the reward obtained via interaction with the environment. The game has four states, each representing a full-stack configuration of the system– {(py, MySQL),(py, PostgreSQL),(Php, MySQL),(Php, PostgreSQL))}. We consider three attacker types– $\mathcal{A}_1$ (database hacker) with 269 actions, $\mathcal{A}_2$ (script kiddie) with 34 actions, and $\mathcal{A}_3$ (mainstream hacker) with 48 actions (detailed CVE-list can be found in the supplementary material).

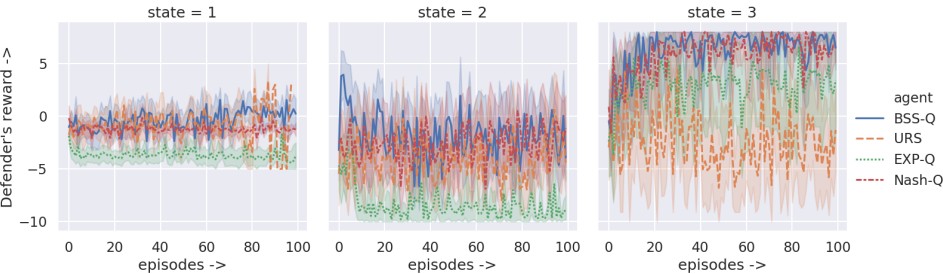

Figure 4: Values in the BSMG-based MTD for IDS placement when using BSS Q-learning (blue) compared to Uniform Random Strategy (orange), EXP-Q learning (green) and Nash Q-learning (red).

The defender's policy in each state is a mixed strategy that directs how to switch to a different configuration, while an attacker type's policy prioritizes attacks that cause maximum damage. In Figure 2, we plot the defender's reward (over 6 trials) in this BSMG for our BSS-Q learning agent and other baselines. In each setting, we use a discount factor of $\gamma = 0.8$, an exploration rate of $\epsilon = 0.1$ (that gradually decays to 0.05) and initiate the agents with a uniform random strategy (except in the case of S-OPT explained as follow). The average time used by the methods for one trial are is shown in Figure 3. We will now explain all the baselines considered.

**Static Movement Policies** These defense policies, evident from its name, are determined up-front (using game parameters provided initially) and do not change. The placebo baseline, used as a sanity check in the context of MTDs, is the Uniform Random Strategy (URS); it selects each action in a state with equal probability (34). Then, we consider the state-agnostic optimal policy (S-OPT) determined by the game-theoretic formulation of the MTD with switching costs (6); this is the state-of-the-art movement policy in this scenario.

| Agents | Time (sec) |
| --- | --- |
| Static | $84.097 \pm 0.157$ |
| B-EXP-Q | $224.827 \pm 1.449$ |
| BSS-Q | $151.127 \pm 13.403$ |
| B-Nash-Q | $> 3600$ |

Figure 3: Run-time of the learning agents for 80 episodes.

**Learning Agents** In (13), the authors leverage adversarial multi-arm bandits in learning policies for an agent in Markov Security Games. While the paper draws inspiration from works in Stackelberg Security Games (4), the EXP Q-learning approach does not consider (1) a strategic adversary that can adapt or (2) uncertainty over attacker types. We adapt their algorithm for BSMGs by ensuring that the update to the sum of rewards is weighed by the attacker type's probability. We call this the Bayesian EXP Q-learning agent (B-EXP-Q). The Bayesian Nash Q-learning (B-Nash-Q) (14), even after we remove the Bayesian update of $\theta_s$, does not scale for this domain and thus, can only be compared against in simple toy domains (a non-Bayesian version is discussed for the other MTD).

In Figure 2, we see that the BSS Q-learning agent outperforms URS in all the states of the BSMG and better than B-EXP-Q and S-OPT in $s_2$ and $s_3$, attaining a reward close to 0, which is the maximum reward possible in this game. In $s_0$ and $s_1$, the B-EXP-Q agent, similar to the BSS-Q agent, converges to an optimal movement strategy. Meanwhile, in $s_3$, the best response of the attacker results in lower rewards for a particular defense action, in turn making this action less probable. As soon as the defender switches to a more promising action, the attacker changes their response to yield a low value for that action. It should be no surprise that defense strategies learned via single-agent RL methods (eg. (13; 15; 16)) are prone to be exploitable against strategic opponents in cyber-security scenarios. The existence of such a cycle makes the learned policy exploitable, resulting in low rewards consistently. In such cases, even the URS yields better rewards because its lack of bias makes it less exploitable. The S-OPT, as described in (6), yields a strategy that moves between two MTD configurations represented by $s_2$ and $s_3$. As such a strategy can never find itself in any other state of the game, to be fair, we ensure that the start state in each episode is uniformly sampled from $s_2, s_3$. Thus S-OPT has no footprint in the states $s_0$ and $s_1$. As stated before, the S-OPT strategy, a resultant of the state-agnostic game-theoretic formulation, is doomed to be sub-optimal. Unsurprisingly, the policy learned by BSS-Q is shown to be better in $s_2$ and $s_3$. A drop in rewards near episode 70 in $s_1$ for BSS-Q can be attributed to the discovery of powerful attacks by two follower types in one trail.

MTD FOR IDS PLACEMENT

We consider the General-Sum Markov Game formulated in (20). As described earlier, this domain has four states, of which, one is a terminal state. As the set of follower types in this game in singleton, the BSMG model boils down to Markov Game. Hence, the vanilla EXP-Q agent can be used (albeit against a rational follower). The URS baseline remains unchanged, S-OPT deems to exist and instead of Bayesian Nash, we can consider the relatively more scalable Nash Q-learning (Nash-Q) agent (35). The rewards obtained by the various agents in the three non-terminal states of the game are plotted in Figure 4. We do not plot the rewards obtained by the model-based inference algorithm in (20) because the policy learnt by the proposed BSS-Q agent eventually converges to an optimal policy at SSE (plotting both results in overlapping graphs after BSS-Q converges, hampering readability). Given the domain has relatively fewer actions, we average the reward over 10 trials; each trial had 100 episodes and with an exploration rate of 0.1 and a discount factor of 0.8.

The BSS Q-learning agent outperforms the two previous baselines– URS and EXP-Q– in at least one of the three states. In this setting, the Stackelberg threat model adversely affects the policy learned by EXP-Q resulting in consistent low rewards for states $s_1$ and $s_2$. While (20) shows that the SSE $\subseteq$ NE for the MG owing to the structure of the defender's strategy sets, we see that Nash-Q performs slightly worse the BSS-Q in state $s_1$. This happens because the existence of multiple Nash Equilibria throws-off Nash-Q from the optimal reward path (36). Similar rewards are observed in all other states.

## 5   DISCUSSION AND RELATED WORK

**The Landscape of Existing Games**   We seek to answer two questions– where does our BSMG fit in and why it is useful. In Figure 5, we graphically situate BSMG in the landscape of existing work. BSMG, as seen in the experiments, can generalize Markov Games (18) (and therefore, MDP). Instead of assuming infinite reasoning capabilities required for Bayesian Nash equilibrium (37), Bayesian Markov Game considers scenarios where players have finite levels of belief about other players(s) (38). On the other hand, Markov Games with Imperfect Information (MGIIs) assumes a Markovian property over the state, the observations, and the joint set of actions; this results in reasoning over opponent types and allows them to decompose Partially Observable Stochastic Games (POSGs) (39) into a set of Bayesian stage-games (40). In BSMGs, the assumption of a pre-specified distribution over attacker types helps us (1) avoid reasoning over the nested belief space, and (2) can be interpreted as the private information of the opponent in MGIIs (and POSGs) as being provided upfront. Thus, BSMG becomes a special case of MGII and BMG with the added semantics of leader-follower interaction. Our assumptions (about modeling imperfect information) helps our game be scalable while providing adequate expressive power in the context of MTDs.

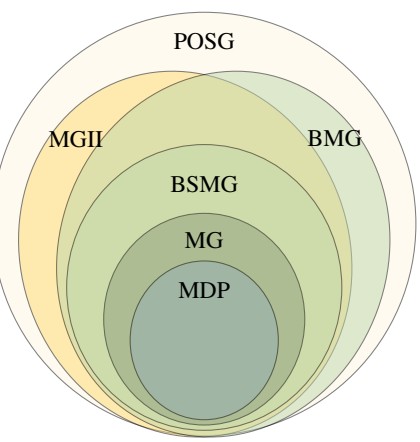

Figure 5: Situating BSMGs in the landscape of game-theoretic models that capture incomplete information.

**Multi-agent Reinforcement Learning in Markov Games**   A standard solution strategy in MGs, when $\tau$ and $U$ are unknown but a simulator is available, is to adapt the Bellman's update used in the single-agent Markov Decision Processes (MDPs) for multi-agent reinforcement learning to learn equilibrium policies (36). In the context of MARL, researchers have investigated different notions of equilibrium. The min-max Q-learning is meaningful when the game has a zero-sum reward structure (41). On the other hand, Nash Q-learning, introduced in (35), has been categorized into two types by (42)– Friend, where the game defined by the Q-values always allows for an optimal joint action profile, and Foe, where the game admits a saddle point solution. The convergence of these algorithms is mostly shown by the fact that Q-values of the states in self-play, given infinite exploration, approach the *correct* Q-values (i.e. calculated Q-values if all the parameters of the game

were provided upfront). The convergence of Nash Q-learning in the context of general-sum games becomes difficult because of the existence of multiple equilibria and the lack of a common incentive or co-ordination amidst agents (36). In correlated equilibrium (CE) Q-learning (43), authors assume the existence of a correlation device accessible to both players. However, the authors show that the learned strategies of the players converge to an uncorrelated equilibrium. On the other hand, in Stackelberg Q-learning (32), there exists a leader-follower paradigm among the players, i.e .the leader's strategy can be observed by a follower before the latter commits to an action. Convergence guarantees in self-play, although popular, become less meaningful when action sets of the players are different (defense configuration *vs.* exploits) and game utilities have a general-sum reward structure.

**Leader-follower scenarios** Researchers have investigated solution concepts in the context of Stochastic games with multiple followers, but do not model incomplete information about them.[2] The interaction is generally modeled as a Semi-Markov Decision Process (44) and improvements consider (1) multiple followers (45), (2) factored state spaces (46), (3) methods based on deep reinforcement learning (47; 48) etc.

**Reinforcement Learning in MTDs** Works that are precursors to our BSS-Q learning approach are the min-max Q-learning (21) in the context of complete information MGs and the Bayesian Nash Q-learning (14) for dynamic placement of sensors. Recent works that model the multi-agent cyber-scenarios as an MDPs (RL in Flip-it games (15)) or POMDPs (RL for MTDs (16)) generate policies that can be exploited by a strategic adversary.

## 6 CONCLUSION

We proposed a Bayesian Stackelberg Markov Game (BSMG) that considers the leader-follower scenario and uncertainty over adversary types in Markov Games. We showed that BSMGs are a unified framework to characterize optimal movement policies in Moving Target Defenses (MTDs) and then proposed a Bayesian Strong Stackelberg Q-learning (BSS-Q) approach to learn robust policies when the rewards and transitions are absent. We showed that BSS-Q learning converges to the SSE of the BSMG. Experiments conducted in two cyber-security scenarios– MTD for web-application and MTD for cloud-networks– showed that policies learned using BSS-Q outperform existing baselines. Supplementary material contains proofs, environment details and additional experiments.

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

## A  APPENDIX – PROOFS

In this section, we present the proofs associated with the lemmas that characterize follower strategies and Strong Stackelberg Equilibrium (SSE) in Bayesian Stackelber Markov Games (BSMGs). We then show that, under specific conditions, the SSE Q-Learning (SSE-Q) converges to the SSE of BSMGs.

**Lemma 1.** For a given policy of the leader/defender in BSMG, every follower/attacker type will have a deterministic policy $\forall s \in S$.

We note that BSMGs adhere to the Markovian nature and thus, a leader's policy is a Markov stationary policy (7). For each follower type, a modified state transition function $\tau'$ that accounts for (1) the original transition $\tau$ and (2) the leader policy $x$ constitutes a Markov Decision Process (MDP) (i.e. $\tau' = \tau \cdot x$). This guarantees that each follower type has a deterministic best-response policy given a leader's policy.

**Corollary 1.** For an SSE policy of the defender, denoted as $x$, each attacker type $A_i$ has a deterministic policy $q_i$. The action $(x, q_1, \ldots q_t)$ denotes the SSE of the BSMG.

We can now extend results known for Markov Games with a single follower type with a singleton response set where the distinction between Strong and Weak SE does not arise (32).

**Lemma 2.** An action profile $(x, q_1, \ldots, q_t)$ that yields the equilibrium values $V_{x,q}^{\mathcal{D}}$ and $V_{x,q}^{\mathcal{A}_i}$ to the players is at SSE of BSMG, *iff* $\forall s \in S$ $(x(s), q_1(s), \ldots, q_t(s))$ is an SSE of the bi-matrix Bayesian game represented by the Q-values $Q_{x,q_i}^{\mathcal{D}}(s), Q_{x,q_i}^{\mathcal{A}_i}(s)$.

First, note that a follower type can have a pure strategy response that corresponds to a Strong Stackelberg Equilibrium for single-stage normal-form games. Given a defender's policy, the attacker solves a linear (reward) maximization that always has a pure strategy in support of an optimal mixed strategy (22). Hence, $q_i(s)$ is a pure-strategy of the bi-matrix game represented by the Q-values if

state $s$. This ensures that the SSE of the BSMG (Corollary 1) and the SSE of the bi-matrix games in each state admit pure-strategy for the individual follower types.

We will now prove the lemma in the forward direction by considering a proof by contradiction. Let us assume that (1) $(x, q_1, \ldots, q_t)$ is an SSE action profile of BSMG with equilibrium values $V_{x,q_i}^p \ \forall p \in P$ but, (2) $\exists s \in S$ for which $(x(s), q_1(s), \ldots, q_t(s))$ is not the SSE of the bi-matrix Bayesian game defined by the Q-values of $s$. If it were so, given that an SSE is bound to exist for the bi-matrix Bayesian game and it yields the highest unique pay-off to the players in a bi-matrix Bayesian game, a player $p$ ($D$ or $A_i$) should switch from their current strategy to an SSE in state $s$. This would clearly yield values higher than $V_{x,q_i}^p(s)$ for that state. This violates (1) because $V_{x,q_i}^p(s)$ was the equilibrium values of the BSMG corresponding to an SSE policy that has a unique optimal value. Thus, (1) implies (2).

A similar proof by contradiction can be constructed for the backward direction. Briefly, if the strategy in the states constitute SSE of the stage game but are not an SSE of the BSMG, it must be possible to switch the strategy in at least one state to yield the higher value guaranteed by the SSE of the BSMG. But if that is the case, the original assumption that the initial strategy in that state is an SSE is contradicted. $\qquad\square$

### A.1 PROPOSITION 1 (CONVERGENCE RESULTS)

Our proof of convergence is inspired from the initial work by (33). Let us call the Q-value update step as a process $\Omega : \mathbf{Q} \to \mathbf{Q}$ where $\mathbf{Q}$ represents the space of Q-values. Formally we can express the update equation for the leader $\mathcal{D}$ as,

$$\Omega(Q^{\mathcal{D},i}(s, a^{\mathcal{D}}, a^{\mathcal{A}_i})) \ = \ U^{\mathcal{D},i}(s, a^{\mathcal{D}}, a^{\mathcal{A}_i}) + \gamma V^{\mathcal{D}}(s_{T+1})$$

Where $V^{\mathcal{D}}$ represents the leader's game value in the Bayesian Stackelberg Game (BSG) defined by Q-values and the distribution over follower types in state $s_{T+1}$. With some abuse of notation, we can drop the arguments $(s, a^{\mathcal{D}}, a^{\mathcal{A}_i})$ for both the functions $Q^{\mathcal{D}}$ and $U^{\mathcal{D}}$ and rewrite the above equation by expanding the value function as follows.

$$\Omega(Q^{\mathcal{D},i}) = U^{\mathcal{D},i} + \gamma V^{\mathcal{D}}(s_{T+1})$$

To prove convergence of the function/operator/process $\Omega$, we need to show the following two conditions hold (as described in (33); the other two conditions mentioned hold trivially in our setting, as it holds in the case of other Q-learning approaches).

(1) The following processes converge to a fixed point.

$$Q_{T+1}^{\mathcal{D},i} \ = \ (1 - \alpha_T) Q_T^{\mathcal{D},i} + \alpha_T \Omega(Q_{sse}^{\mathcal{D},i}) \ \forall \, i$$
$$Q_{T+1}^{\mathcal{A}_i} \ = \ (1 - \alpha_T) Q_T^{\mathcal{A}_i} + \alpha_T \Omega(Q_{sse}^{\mathcal{A}_i}) \ \forall \, i$$

where $Q_{sse}$ represents the Q-values at SSE of the BSMG.

(2) The process $\Omega$ is a real contraction operator.

$$||\Omega(Q) - \Omega(\bar{Q})|| \le a||Q - \bar{Q}|| \quad \forall Q, \bar{Q} \in \mathbf{Q}$$

where $0 < a < 1$ and $|| \cdot ||$ denotes the supremum operator over the vector space $Q$.

To prove the conditions in (1), we leverage the Condition Averaging Lemma stated in (33) and thus, have to show that,

$$Q_{sse}^{\mathcal{D},i} = \mathbb{E}[\Omega(Q_{sse}^{\mathcal{D},i})] \quad , \quad Q_{sse}^{\mathcal{A}_i} = \mathbb{E}[\Omega(Q_{sse}^{\mathcal{A}_i})] \ \forall \, i$$

where the expectation is over the states reached. To show this, we first expand the right hand side of the equation and showing that this expansion is equal to the left hand side.

$$
\begin{aligned}
\mathbb{E}[\Omega(Q_{sse}^{\mathcal{D},i})] \ &= \ U^{\mathcal{D}} + \gamma \, \mathbb{E}[V^{\mathcal{D}}(s')] \\
&= \ U^{\mathcal{D}} + \gamma \, \mathbb{E}[V_{sse}^{\mathcal{D}}(s')] \\
&= \ U^{\mathcal{D}} + \gamma \sum_{s'} \tau(s'|s, \sigma) * V_{sse}^{\mathcal{D}}(s') \\
&= \ Q_{sse}^{\mathcal{D},i}
\end{aligned}
$$

where $V_{sse}^{\mathcal{D}}$ indicates the game value of the defender at SSE. $V^{\mathcal{D}} = V_{sse}^{\mathcal{D}}$ because the Q-matrices in all state represent the value at SSE. The final equality is a result of the Bellman equation for multi-agent settings. It is easy to see that the same line of reasoning holds for all all follower types.

To prove the condition in (2), we will first expand the Left Hand Side (LHS) followed by the expansion of the Right Hand Side (RHS). Then we will show that a stricter case of the inequality is satisfied. We first show this for the follower and then, for the leader.

$$
\begin{aligned}
& ||\Omega(Q^{\mathcal{A}_i}) - \Omega(\bar{Q}^{\mathcal{A}_i})|| \\
= & \max_s \left( \Omega(Q^{\mathcal{A}_i}(s, x, R(x))) - \Omega(\bar{Q}^{\mathcal{A}_i}i(s, x, R(x))) \right) \\
= & \gamma \max_s \left( V^{\mathcal{A}_i}(s') - \bar{V}^{\mathcal{A}_i}(s') \right) \\
= & \gamma \max_s \left( \max_x Q^{\mathcal{A}_i}(s', x, R^{\mathcal{A}_i}(x)) - \max_x \bar{Q}^{\mathcal{A}_i}(s', x, R^{\mathcal{A}_i}(x)) \right) \\
= & \gamma \left( \max_x Q^{\mathcal{A}_i}(s', x, R^{\mathcal{A}_i}(x)) - \max_x \bar{Q}^{\mathcal{A}_i}(s', x, R^{\mathcal{A}_i}(x)) \right)
\end{aligned}
\tag{1}
$$

The first equality is based on the use of the *supremum* operator. Given that the max occurs for some $s$, without loss of generality, we can assume this state is $s$ going to state $s'$. In a similar way, we can expand the RHS for the follower.

$$
\begin{aligned}
& a||Q^{\mathcal{A}_i} - \bar{Q}^{\mathcal{A}_i}|| \\
= & a \max_s \max_x \max_q \left( Q^{\mathcal{A}_i}(s, x, q) - \bar{Q}^{\mathcal{A}_i}(s, x, q) \right) \\
\geq & a \max_x \max_q \left( Q^{\mathcal{A}_i}(s', x, q) - \bar{Q}^{\mathcal{A}_i}(s', x, q) \right) \\
\geq & a \max_x \left( Q^{\mathcal{A}_i}(s', x, R^{\mathcal{A}_i}(x)) - \bar{Q}^{\mathcal{A}_i}(s', x, R^{\mathcal{A}_i}(x)) \right)
\end{aligned}
\tag{2}
$$

Note that we now have a stricter version of the RHS. If we can now show that Equation 1 $\leq$ Equation 2, then we can prove condition (2) holds for the Q-values of all follower types. Given $0 \leq \gamma < 1$, we can consider $a = \gamma$. Now we have,

$$
\begin{aligned}
& \gamma \left( \max_x Q^{\mathcal{A}_i}(s', x, R^{\mathcal{A}_i}(x)) - \max_x \bar{Q}^{\mathcal{A}_i}(s', x, R^{\mathcal{A}_i}(x)) \right) \\
= & a \left( \max_x Q^{\mathcal{A}_i}(s', x, R^{\mathcal{A}_i}(x)) - \max_x \bar{Q}^{\mathcal{A}_i}(s', x, R^{\mathcal{A}_i}(x)) \right) \\
\leq & a \max_x \left( Q^{\mathcal{A}_i}(s', x, R^{\mathcal{A}_i}(x)) - \bar{Q}^{\mathcal{A}_i}(s', x, R^{\mathcal{A}_i}(x)) \right) \\
\leq & a||Q^{\mathcal{A}_i} - \bar{Q}^{\mathcal{A}_i}||
\end{aligned}
$$

The first inequality holds because in the second step, one can select two different $x$-s to minimize the difference while in the third step, one is constrained to select the same $x$ for both the Q-values.

Now, we show that $\Omega$ is also a contraction operator for the Q-values of the defender. Showing the property holds is difficult to show for individual follower types because of the Bayesian nature of the game. It is possible to show this for a transformed attacker conjured using the Harsanyi transformation (49). In this setting, the single attacker type has actions that are cross product of the action of all other players and the utilities of the Q-value matrix is the expected Q-value over the original attacker types. Given this single attacker type, we use $Q^{\mathcal{D}}$ to denote the Q-values against this new transformed attacker. Given the solver we are using in our BSS Q-learning approach in calculating SSE of the BSG stage games is equivalent to the SSE of this transformed game (22), showing $\Omega$ is a contraction operator for $Q^{\mathcal{D}}$ is sufficient to show convergence. We use $\mathcal{A}$ in the superscripts to denote value for this transformed attacker type.

$$
\begin{aligned}
& ||\Omega(Q^{\mathcal{D}}) - \Omega(\bar{Q}^{\mathcal{D}})|| \\
=\ & \max_s \left( \Omega(Q^{\mathcal{D}}(s, x, R^{\mathcal{A}}(x))) - \Omega(\bar{Q}^{\mathcal{D}}(s, x, R^{\mathcal{A}}(x))) \right) \\
=\ & \gamma \max_s \left( V^{\mathcal{D}}(s') - \bar{V}^{\mathcal{D}}(s') \right) \\
=\ & \gamma \max_s \left( \max_x \sum_i \theta^i(s') Q^{\mathcal{D},i}(s', x, R^{\mathcal{D},i}(x)) - \max_x \sum_i \theta^i(s') \bar{Q}^{\mathcal{D},i}(s', x, R^{\mathcal{D},i}(x)) \right) \\
=\ & \gamma \left( \max_x \sum_i \theta^i(s') Q^{\mathcal{D},i}(s', x, R^{\mathcal{D},i}(x)) - \max_x \sum_i \theta^i(s') \bar{Q}^{\mathcal{D},i}(s', x, R^{\mathcal{D},i}(x)) \right) \\
=\ & a \left( \max_x \sum_i \theta^i(s') Q^{\mathcal{D},i}(s', x, R^{\mathcal{D},i}(x)) - \max_x \sum_i \theta^i(s') \bar{Q}^{\mathcal{D},i}(s', x, R^{\mathcal{D},i}(x)) \right) \\
\leq\ & a \max_x \left( \sum_i \theta^i(s') Q^{\mathcal{D},i}(s', x, R^{\mathcal{D},i}(x)) - \sum_i \theta^i(s') \bar{Q}^{\mathcal{D},i}(s', x, R^{\mathcal{D},i}(x)) \right) \\
\leq\ & a \max_s \max_x \left( \sum_i \theta^i(s) Q^{\mathcal{D},i}(s, x, R^{\mathcal{D},i}(x)) - \sum_i \theta^i(s) \bar{Q}^{\mathcal{D},i}(s, x, R^{\mathcal{D},i}(x)) \right) \\
\leq\ & a \max_s \max_x \Pi \max_{q^{\mathcal{A}_i}} \left( \sum_i \theta^i(s) Q^{\mathcal{D},i}(s, x, q^{\mathcal{A}_i}) - \sum_i \theta^i(s) \bar{Q}^{\mathcal{D},i}(s, x, q^{\mathcal{A}_i}) \right) \\
\leq\ & a \max_s \max_x \max_{q^{\mathcal{A}}} \left( Q^{\mathcal{D}}(s, x, q^{\mathcal{A}}) - \bar{Q}^{\mathcal{D}}(s, x, q^{\mathcal{A}}) \right) \\
=\ & a ||Q^{\mathcal{D}} - \bar{Q}^{\mathcal{D}}||
\end{aligned}
$$

If we were now to consider $\bar{Q} = Q_{sse}$ for all the player and player types, then the Q-values learned by our method will approach $Q_{sse}$. While this completes our convergence proof, we note that convergence rate depends on two factors. First, Selecting randomly among best-responses, even if multiple exist, for the follower results in slower convergence. Selecting consistently in some order (eg. first after sorting the response set) results in faster convergence. Note that random selection does not cause issues beyond slowing down convergence because for each follower type, given a leader's strategy, regardless of the best response strategy selected, the game value for both players remains the same due to nature of SSE (25; 26). Second, a similar line of reasoning for the defender concludes that a pre-defined selection mechanism can result in faster convergence.

## B    APPENDIX – ENVIRONMENT DESCRIPTION

In this section, we first describe the Open-AI style game-simulator interface that can be used by the learning agents. We then briefly describe how some of the simulator functionalities are provided for the domains and future directions in this regard.

```
def get_states():
 ...
 return []
"""
@Input
 None
@Output
 Returns a list (essentially a set) consisting of states in the game.
"""

def get_start_state():
 ...
 return s
"""
@Input
 None
@Output
 A start state s ∈ start_S ⊆ S denoting the start state for an episode.
"""
```

```
20
21  def get_actions():
22    ...
23    (return [[ [], [], ... ], []])
24    """
25    @Input
26      None
27    @Output
28      A list with the first element representing attacker actions and the
          second element representing defender actions. The first list can be
          further decomposed in to set of lists representing actions for each
          attacker/follower type.
29    """
30
31  def is_end(s):
32    ...
33    return True/False
34    """
35    @Input
36      A state s ∈ S.
37    @Output
38      AssertionError if s ∉ S.
39      True if the input state s ∈ end_S ⊆ S
40      False otherwise
41    """
42
43  def act(s, a_D, a_A, θ):
44    ...
45    return R_D, R_A, s_{t+1}
46    """
47    @Input
48      A state s ∈ S, defender's action a_D, attacker/follower's action a_A, the
          attacker type θ
49    @Output
50      AssertionError if s ∉ S, a_D (or a_A) is not a defender (or attacker type θ
          's) action.
51      R_D -- Defender's utility
52      R_A -- Attacker's utility
53      s_{t+1} -- Next state
54    """
```

### B.1 MOVING TARGET DEFENSE FOR WEB-APPLICATIONS

As mentioned earlier, we leverage the fully specified game domain from (6) to build the simulator. In the context of the simulator functions, the `get_state` method returns four states of the system. Each state of the defender constitutes choosing an implementation language (Php or python) and a database technology (SQL or PostgreSQL) that can be used to host the web-application.

The `get_start_state` method of the game simulator returns on of the four states at random, implying the system can start in any one of the four configurations. We allow a user to override the global variable that describes the set of start states because, in the context of certain baseline strategies such as BSG formulation (6), they consider only a sub-set of the MTD configurations. Thus, a strategy that places zero probability of switching to a configuration can new start or find itself in the configuration.

The `get_actions` method returns the set of actions available to the attacker types and the defender. Similar to the original game designed in (6), we pair up an attacker types expertise level and a set of technologies it has expertise in to the Common Vulnerabilities and Exploits (CVEs) mined from the NVD database to define their attack set. For the defender, the four configurations it can choose constitute the attack set.

The `is_end` always returns False in this setting as the game has not terminal state and the goal of the defender is to continuously keep moving the system. While we can stop when policy converges for

Figure 6: Comparing BBS-Q with URS (orange) and EXP-Q learning (green) on MTD for web-applications with switching costs incorporated in the transition function of BSMGs.

our proposed BSS-Q algorithm, for the other algorithm, owing to the lack of convergence guarantees, we run it for a predefined number of episodes.

The `act` method considers the action of the defender and the attacker's actions to determine the impact. Ideally, this can be done by deploying the system on a new configuration and then sending out an actual request to the web-service with the attack folder as part of a request. Then, depending on the return, decide if the attack was successful or not. While we seek to generate a class of attacks that can represent the 300+ CVEs used in the domain, this was out of scope for this work. Further, the attack success or failure might only give a binary indication of the rewards fro the attacker, which constitutes a sparse signal and treats impactful and trivial attacks under the same umbrella. To address this, we leverage the Common Vulnerability Scoring Service (CVSS) and use the Impact score as the attacker type reward if the chosen attack is expected to work on the defense configuration being deployed. We use the current state of the system and the defender action to compute the cost of the movement. Ideally, we want to determine this by running a system with multiple virtual machines– one hosting the current configuration and the other bringing up the next configuration (determined by $a_{\mathcal{D}}$). The time taken in bringing up the new configuration and then the amount of packets dropped or the extra resources used in the switching should all be part of the switching costs. While one can come up with an elaborate procedure to do so, this is beyond the present scope of our paper and we consider the costs calculated based on configuration-based similarity (6)– configurations that are more dissimilar incur higher switching costs.

### B.1.1 REAL-TIME DECISION BASED ON SWITCHING COST THRESHOLD

We consider a different perspective on switching costs that is possible to represent using BSMG but cannot be captured by existing work in (6). In this setting, we do not capture the switching cost as part of the reward metric, but use it to guide the transition dynamics of the underlying environment. Precisely, when an `act` API is called, we use the defender's action $a_{\mathcal{D}}$ to perform the switch. While performing the switch action, if we observe a drop in the successful processing of packets or an increase in the response latency, calibrated by a threshold, we abort the move action and stay in the same state. This introduces stochasticity in the underlying environment, which, as per the domain in (6), was initially deterministic. We use the existing switching cost to guide the transition dynamics, i.e. expensive switches have a higher probability of failing to execute the switch and thus, remain in the same state.

**Experimental Results** In Figure 6, we plot the rewards obtained by the proposed BSS Q-learning agent in comparison to the baselines described in the paper– namely the Uniform Random Strategy (URS) and the adversarial multi-arm bandit based EXP-Q learning agent. We ignore the state-agnostic optimal policy in this setting because it only expects to see itself in two configuration of the system and thus, becomes sub-optimal in this setting given the stochastic nature of the environment (that it cannot even model). We use a discount factor of $0.8$, a exploration rate of $0.15$ (that decays gradually to $0.05$ and a learning rate of $0.06$.

Similar to our results in the scenario described in the paper, BSS-Q gathers higher reward, over six trails, than URS. In this setting, it performs better than EXP-Q in three states $s_0$, $s_2$ and $s_3$ (the margin of improvement being significantly better in $s_0$ and $s_2$).

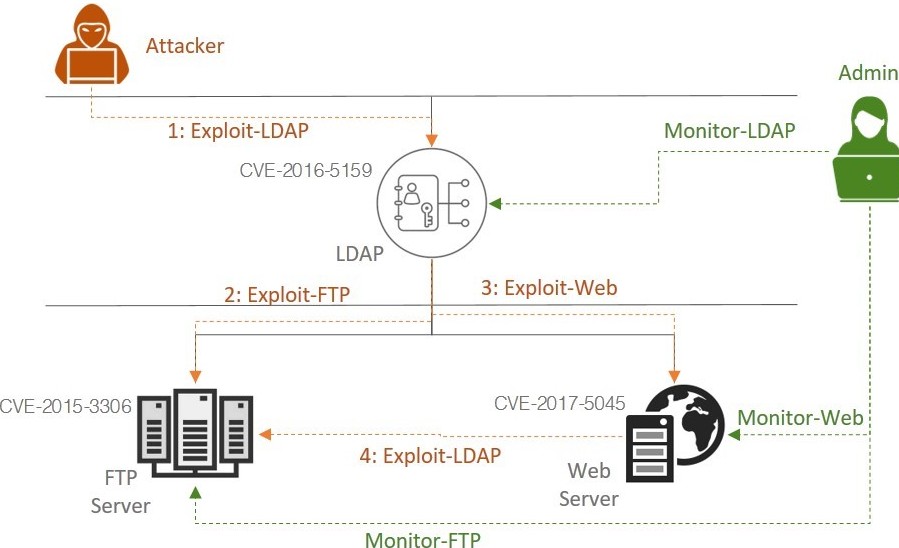

Figure 7: An example cloud system highlighting its network structure, the attacker and defender (admin) agents and the possible attacks and monitoring mechanisms (20).

## B.2 MOVING TARGET DEFENSE FOR PLACEMENT OF INTRUSION DETECTION SYSTEMS IN CLOUD NETWORKS

Before describing the design of the simulator, we provide the underlying cloud network, scenario, derived from the modeling in (20), in Figure 7. This scenario, via two transformation steps– first, to an attack graphs and then to a Markov Game– can be used to build our game simulator. For details of this process, we refer the reader to look at (20).

The get_start_state method of the game simulator returns the single start state that represents the case where an attacker has user access to the LDAP server on the public facing interface of the network system.

The get_actions returns the set of attack actions, discovered using vulnerability scanners on the cloud system and a part of the attack graph representation in (20), that are possible for an attacker to execute in a given state of the cloud network system. The action set for the defender indicates the set of Intrusion detection systems they can place and a no-op action implying that they may not choose to place an IDS system at all.

The is_end returns a single state of the BSMG. This state represents the condition where the attacker has administrator access on the file server.

The act method considers the action of the defender and the attacker's actions to determine if the attack action is detected. Ideally, we want to play the defender's strategy of placing IDS system and then execute the attack action chosen by the attacker. While this needs an entire cloud network setup with Virtual Machines (VMs), we use similar resources used in (20) to determine the utilities and the transition. If the IDS placed is able to detect the exploit, we return a reward proportional to the effort required by the attacker in executing the attack action. This, similar to (20), is determined using the exploitability score of the CVE determined from the Common Vulnerability Scoring System (CVSS). If the IDS systems fails to detect the attack, then the attacker has an impact proportional to the base score obtained from CVSS which considers both the impact and the complexity of the attack vector. Futher, the game transitions into a new state where the attacker has either escalated privileges on the same VM (or physical server) or gets access to a new VM on the cloud. The defender besides the impact score, which represents the security impact, considers the impact on network bandwidth if they deploy a Network Intrusion Detection System (NIDS) and impact on memory or CPU resources if they deploy a Host-based Intrusion Detection System (HIDS). We use the scaling used in (20) to come up with a single utility value for the defender.

