# OpenReview forum: "Learning Movement Strategies for Moving Target Defense"
_ICLR.cc/2021/Conference — Reject_

### Official Review · AnonReviewer2 · 2020-10-23
**ICLR 2021 Conference Paper966 AnonReviewer2**

**Rating:** 4
**Confidence:** 2

**Review:**

Summary:

This paper studies the problem of learning how to adapt the defense methods in the domain of cybersecurity. The paper proposes a new model called Bayesian Stackelberg Markov Games (BSMG) to capture the uncertainty of the attacker's types as well as their strategic behaviors. The authors design Bayesian Strong Stackelberg Q-learning that can converge to the optimal movement policy for BSMG. The empirical studies verify the support the theoretical results.

Detailed Comments:

The empirical results give evidence that the proposed method is effective in practice. However, the reviewer had a hard time to understand the model of BSMGs and how the attackers behave in this model.

1. Given the definition of \Theta, i.e., the probability distribution of the attackers, it seems that the attacker is freshly drawn for each round according to \theta_k if the state is s_k, and the distribution is independent of the game history. However, the state transition function \gamma depends on the attacker's type and action, which is very confusing. If the attacker's type is redrawn in each round, then how should the attackers reason about their strategies for the current round? Are they myopic? But this contradicts with the description of Algorithm 1 in which the Q-value of the attacker's are computed, which implies that the attackers care about the future. Could you clarify on this?

2. Is it necessary to assume that the discounting factor for all attackers are the same? Does the result depend on this assumption?

---

> ### Author Response · Authors · 2020-11-22
> **Clarification about myopic vs. far-sighted attacker types and impact of heterogeneous discount factors.**
>
> We thank the reviewer for their insightful questions and giving us the opportunity to clarify some of the assumptions that may not have come across properly due to space restrictions. We clarify them here and try to incorporate them in the rebuttal version of our paper.
>
> **[Myopic vs Far-sighted attackers]** Note that in algorithm 1, an attacker type’s Q-value is updated based only on the value function of that attacker type; thus the attacker is not clearly myopic. Note that an attacker type’s value function doesn’t have to be updated right away in the same episode’ and can be done across multiple episodes. An easy way to understand this is to consider a setting where an attacker type is sampled for an episode and we let the defender and this attacker type interact till an end state is reached. In the next episode, we again sample an attacker type and consider the interaction (episode-level sampling as opposed to state-level sampling). In that way, the defender’s policy overfits to one attacker in one episode, then does terribly bad on the next episode if the sampled attacker type is different. Over many episodes, it eventually learns a robust policy that works best and is at equilibrium against the distribution of attackers. An alternative but faster way to ensure convergence is to bring the notion of sampling to the state-level as opposed to the episode-level. Given the value function is used across the episodes and an attacker type only uses its value function to update its Q-values, the attacker type still remains far-sighted into the future. The overfitting of the defender’s policy to a particular attacker across episodes doesn’t happen in our setting and helps in (1)  faster convergence and (2) has a lesser reward variance.
>
> **[Discounting Factor may vary for the players]** This is possible in the real world where some attacker types may care about short-term objectives (bragging rights on exploiting the first vulnerability of an enterprise network) vs. hacking the database deep-rooted in the cloud network. The convergence proofs still holds in this case but the rates of convergence will start to vary. For example, if an attacker has a high discount factor, the defender may feel complacent that it has a good policy after a few episodes till this attacker finds a high end-goal. Further, if this attacker type has low probability, they may take a long time to even uncover this high rewarding goal that will affect their policy in states far away from the goal; this delays the convergence.

---

### Official Review · AnonReviewer1 · 2020-10-28
**Review of "Learning Movement Strategies for Moving Target Defense"**

**Rating:** 4
**Confidence:** 4

**Review:**

#########################
PAPER SUMMARY
#########################

This paper proposes the game-theoretic model of Bayesian Stackelberg Markov Games (BSMGs), a generalization of Markov games, as a formalism for studying Moving Target Defense (MTD) systems, a type of defender-attacker game with applications to cybersecurity. An algorithm for finding the Stackelberg equilibrium in BSMGs, called Bayesian Strong Stackelberg Q-Learning (BSS-Q) is proposed, and an OpenAI Gym-style environment for testing the derived policies in particular MTD settings is introduced, which allows for empirical evaluation of the policies' effectiveness. The paper then shows experimental results supporting the BSS-Q algorithm's success at finding the Strong Stackelberg Equilibrium of BSMGs.

#########################
STRONG POINTS
#########################

- Unifying reinforcement learning with leader-follower games is an interesting direction for research.
- The introduction of new learning environments for these settings is itself a potentially valuable contribution.
- The inclusion of the parameters used for the experiments in Section 4 (i.e., discount rate, etc.) aids reproducibility.

#########################
WEAK POINTS
#########################

- The lack of code with the submission doesn't allow for independent verification of the experiments, or for examining the learning environments that have been introduced.
- The sensitivity of the experimental results to choice of parameters is not included. Which choices of parameters mattered and why?
- The discussion in paragraph 3 of page 5 about solving the Bayesian Stackelberg game is unsatisfying; it appears that BSS-Q can only tractably find a solution when the domain is relatively small. Given that the paper claims that a MILP formulation allows for this to be solved effectively in the test domains, then there should be a more detailed discussion of why the test domains presented here are broadly representative of the types of problems that BSS-Q would be expected to be used for solving.
- The experimental results are not presented with sufficient clarity (see the "Questions for Authors" below).

#########################
DECISION RECOMMENDATION
#########################

I recommend rejecting the paper, because I believe that the contributions are not sufficiently broad as to warrant acceptance. In addition, the experimental results are not described in sufficient detail to give confidence about their significance.

#########################
QUESTIONS FOR AUTHORS
#########################

- Can the authors clarify what is meant on page 16, line 1 by "borrowing the game domain": which parts of the cited framework does the system presented in this paper reuse and what has been added?
- What does training of an agent in the system look like? Is learning taking place? It appears, from page 6, line -17, that there is a decay of exploration rate. But what is the training process? Isn't the difficult part of the learning process already handled by the Bayesian Stackelberg game solver?
- The number of trials used in the experiments is inconsistent. Why are 6 trials used in Figure 2 (MTD for Web Applications) while 10 trials are used in Figure 4 (MTD for IDS Placement)? What exactly is a "trial" here - is it a training run, a test run with a trained agent, etc.? If a trial corresponds to a training run, then 100 episodes seems like far too few for agents to learn (at least that is the the case in most other RL domains).
- For each algorithm, Figures 2 and 4 show very similar rewards across the episodes. Does this mean that no learning is taking place for any of the algorithms? If these are test results (i.e., the agents have been trained using these algorithms), then what did the training process look like?
- What does the "time taken by the agents" (Figure 3) signify?
- The choice of baselines here seems to be too weak; as the paper says in paragraph 2 of page 7, the poor performance of baselines is expected, since they are not modeling adversaries at all.

#########################
ADDITIONAL FEEDBACK
#########################

Style suggestions:
- In Figure 1, move the numbers in the graphs on the right-hand side to outside the chart when that part of the chart is too small to contain the numbers.
- Page 13, line -2: The PDF links for Equation 1 and 2 don't appear to be correct; also (1) and (2) are overloaded in this proof, referring both to the conditions for convergence near the top of page 13 and to Equations (1) and (2). This should be clarified for easier reading and for removing ambiguity.

Typographical errors:
- Page 2, line -4: "Bayesian Stackelberg Games (22; 6) is" should be "Bayesian Stackelberg Games (22; 6) are"
- Page 3, line 1: "extends" should be "extend"
- Page 3, line 24: a word like "domains" is missing after "physical (22) and cyber-security (6)"
- Page 4, line 15: "can is better" should be "is better"
- Page 4, lines -5 and -4: "the goal" is repeated twice; one of these should be removed
- Page 7, line -6: "throws of Nash-Q" should be rephrased
- Page 8, line 31: there should be no hyphen in "multiple followers"
- Page 13, line -2: "Note the" should be "Note that"
- Page 17, line 9: "propose" should be "proposed"

#########################
POST-REBUTTAL UPDATE
#########################

Thank you to the authors for your detailed responses and for uploading your code. (Minor point: your README assumes that the GNOME desktop environment is being used; you may want to make the instructions platform-independent.) My main concerns about the learning process described in the paper remain. The authors indicate in their response that "it is difficult to quantify which is the most challenging part of the learning process". This makes it much more difficult to reason about whether the learning process is primarily about using the Bayesian Stackelberg game solver, and whether the interaction with the environment (given the limited number of trials) provides limited benefit.

---

> ### Author Response · Authors · 2020-11-22
> **Clarification and response to concerns raised and uploading code**
>
> We thank the reviewer for their comments and are glad that they found our research on marrying advanced game-theoretic concepts to reinforcement learning as a promising research direction. Further, we are happy to see that moving beyond multi-player games that have homogeneous action set for all players (allowing for self-play) and environments whose connection to the real-world is questionable, they appreciate the use of (cyber-security) domains that are closer to the real-world. We will include our code in supplementary material to improve reproducibility. We now rebut some of the points made in the review.
>
> **[Code Submission]** We apologize for our negligence to submit the code earlier; as stated we will upload our code as a part of the rebuttal version submission.
>
> **[Sensitivity to Choice of Game Parameters]** We discuss the effect of the game parameters that we either expect as input or can control on our algorithm.
> - Discount Factor -- We observed that a lower discount factor leads to faster convergence on average. A lower discount factor ensured that newly discovered paths did not substantially affect the value learned for a state.
> - Learning rate and Exploration rate -- While a smaller learning rate leads to slower convergence, too high a value quickly derails the agent to policies far away from the equilibrium and often biased towards certain actions-- it can take a lot of time for the attacker to find a good action that reduces this bias. Increasing the exploration rate often helps when the learning rate is high because defaulting to the URS policy helps the players sample the less biased actions more often. At the same time, too high an exploration rate can reduce the rate of convergence to a Strong Stackelberg Equilibrium.
>
> **[Scalability of BSS-Q]** The Bayesian Stackelberg Game solvers do indeed have challenges when it comes to scalability. In the context of web-application MTD, a defender cannot be expected to build thousands of code-bases to support a web-application. When we move to an application where thousands of pure-strategies exist for the defender, one can simply replace the BSG solver with improvements made along these lines (although certain assumptions about the reward value have to be made) [1]. In the context of MTD for IDS-placement, the distribution over attacker types goes away making the inference much faster (polynomial time). While the strategy inference can simply scale to larger instances, the sample complexity would increase significantly.
>
> [1] Sinha, A., Nguyen, T. H., Kar, D., Brown, M., Tambe, M., & Jiang, A. X. (2015). From physical security to cybersecurity. Journal of Cybersecurity, 1(1), 19-35.
> [Borrowing the Game Domain]
>
> **[Training an agent]** In this work, similar to existing works in RL, the agent repeatedly interacts with the environment to learn a (provably) high value yielding strategy. In the context of a multi-agent set-up, the learned strategy is at equilibrium. It is difficult to quantify which is the most challenging part of the learning process but, similar to our work, existing Q-learning approaches that seek to learn the Bayesian Nash, Nash, Correlated or mix-max equilibrium also use some approach to discover the equilibrium policies during the learning phase.
>
> **[6 and 10 trials]** The web-application MTD has a huge action set for the attacker (approx 300 actions) in each state whereas, for the IDS-placement MTD, it is relatively small. Due to limited commute power and time constraints, we choose to execute less number of trails for the first domain and a larger number of trails for the second domain.
>
> **[Time taken by the agents]** The time taken by the agents indicates the cpu-time taken by each of the learners to run for 80 episodes (we will change the figure caption to indicate this).
>
> **[Weak baselines]** Existing baselines make stronger assumptions about the adversary (eg. they consider fictitious play) and hence perform worse than ours when faced with a strategic adversary. We still try our best to adapt these methods to our more complex setting for fair comparison and even compare against model-based inference approaches that start with complete knowledge of the domain. For other approaches that relax this assumption, either they are not scalable in our domains (Bayesian Nash-Q, POMDP solvers) or perform comparably to ours (Nash-Q) due to inherent problem structure (in the case of IDS-placement MTD). We believe other baselines (which can be adapted to our setting that we may not be aware of) would be at a disadvantage w.r.t. either scalability or the optimal reward value they converge to (given our approach is guaranteed to converge to an optimal policy at equilibrium).

---

> > ### Comment · AnonReviewer1 · 2020-11-22
> > **Definition of trial**
> >
> > Thank you for your responses. Could you please clarify what you mean by a trial (as in 6 trials vs. 10 trials)?

---

> > > ### Author Response · Authors · 2020-11-23
> > > **Clarification of trials**
> > >
> > > In our setting, we re-start the learning agent 6 times for Web-application MTD and 10 times for MTD for Intrusion Detection System (IDS) placement and plot the mean and variance across the trails.
> > >
> > > In each trial, the agent starts with uniform random strategy but due to a new random seed, the sampling of attacker types and the sampling of next state as per the transition function vary, in turn affecting the convergence rates and the player's value across episodes. Hence, an average value along with its variance, as is common in reinforcement learning, is plotted.
> > >
> > > We also note that we do not use a parameterized (esp. Deep Neural Network based) actor or critic network and thus, the number of episodes (multiple states can be visited till a goal/terminal state is reached in each episode) in each trail is sufficient for convergence.

---

### Official Review · AnonReviewer4 · 2020-10-28
**An interesting application of MDPs to cybersecurity but not introducing enough novelty for the field.**

**Rating:** 5
**Confidence:** 3

**Review:**

In this paper, the authors model a problem of responding to an attacker in a Stackelberg bayesian setting with an MDP. The authors provide a Q-learning-like solution to the problem, its convergence to a Stackelberg equilibrium asymptotically, and some experiments to show the performance of the proposed method.

I think that the pros of this paper are the formal and precise characterization of the analysed problem as an MDP with adversaries, but, in my opinion, this does not constitute enough novelty to be published at ICLR. Moreover, I have some doubts also on the significance of the provided experiments.

The proof of Proposition 1 is key in the results provided by the paper. I think that its proof should be moved to the main paper and, due to the fact that it is not easy to follow, to be revised to improve readability. Moreover, I would like you to state explicitly the difference in the proof w.r.t. the one present in (48).

The experimental evidence you provided does is not statistically significant. Even if the expected value of the different states in the settings you tackled is larger for the proposed method, the confidence intervals do not provide enough evidence that the proposed method is performing better than the baseline. This dramatically compromises the strength of the experimental results you provided.

I think that a strong assumption of the proposed framework is the knowledge of the attackers' distributions. Indeed, usually one has only a little information about the behaviour of the attackers. Do you think it is possible to extend what you proposed also to a setting in which the attackers' distribution is unknown?

Do you think it is possible to evaluate also the loss due to lack of information (regret) in your setting?


Minor:
section 5 -> Section 5
eg. -> e.g.,
algorithm 1 -> Algorithm 1
can’t -> cannot

You should proofread the appendix and check it for errors (e.g., "0 \geq \gamma < 1")

---------------------------------------------------------------------------------------------------------------------
After rebuttals:
The authors made put a significant effort to improve the submission, but I am still non convinced by the experimental results they are presenting. For instance, in Figure 4 there is no way of distinguish between BSS and Nash-Q. I suggest you to increment the repetitions of the experiment to highlight the improvement of your method over the literature ones.

---

> ### Author Response · Authors · 2020-11-22
> **Addressing concerns about novelty, knowledge about attacker type distribution, results, and clarification about convergence proofs**
>
> We thank the reviewer for their comments. We aim for a precise and formal characterization of the cyber-scenario and first-step to learn movement strategies (that give guarantees); we are happy that it came across that way. We now rebut some of the points raised in the review comments.
>
> **[Novelty]** We argue that the contribution of this paper is novel in several aspects. First, while several works have looked at modeling partial observability (POSGs, MGIIs, etc. described in Sec 5 on related work), the one-sided partial observability in cybersecurity provides us an opportunity for tighter modeling that improves the scalability of such methods. Second, proofs of convergence for learning algorithms are rare in partial observability settings and most existing works settle for a better-than-baseline approach. Third, for proactive defenses in cybersecurity, bring the notion of learning policies, and provide an approach for model-free learning alongside gym-like environments for agents to learn. In the latter regard, most existing environments designed for multi-agent RL are toy-environments and often allow self-play as the actions of the different players are the same. Our proposed problem and approach brings more realistic scenarios to the table. (Also, see our response to Reviewer 3.)
>
> **[Convergence Proof]** The convergence proof, aka proposition 1, is quite long and difficult to move to the main body of the paper without affecting the clarity of the problem setup, the proposed model, or the proposed approach. Hence, we will include a proof sketch in the rebuttal version of the paper. The proof that $\Omega$ is a contraction operator for the defender’s Q-value is different from (48); as stated in the last part of the proof (page 14, paragraph 2), it is more challenging due to the presence of multiple follower types and a probability distribution over them given the defender’s partial observability.
>
> **[Statistically Significant Results]** For the web-application MTD, The learned policy in all the states is significantly better than a uniform random strategy, better than the model-based inference strategy in two states (note that the latter is only defined these two stages), and better than EXP-Q in state 3. For the IDS-placement MTD, our learned policy is significantly better than EXP-Q in all three states and better than URS in state 2 (in the two other states, policies in the neighborhood of URS constitute an SSE, and thus, URS yields relatively high rewards). In earlier work (35), the authors show that SSE is also a Nash Equilibrium for the particular game, and thus, a Nash-based Q-learning mechanism yields similar rewards to our algorithm.
>
> **[Knowledge of Attacker Type Distribution]** In cybersecurity settings, a defender often makes assumptions about this probability, and thus, this is not an unrealistic assumption. For example, if a system administrator believes it will always be attacked by a script kiddie, it will not deploy security patches for complex vulnerabilities. On the other hand, if it feels only a database-hacker attacks it, it does not need to defend against any vulnerability unrelated to a database. Thus, any sort of security deployment reasons about a distribution over attack types. Our framework, given this type, produces the best policy for the defender. A simple way to extend this to an unknown distribution over attacker types to incorporate the attacker as a part of the environment; the defender, via interaction, will learn a good policy over the ‘average’ attacker type. Unfortunately, creating such an environment where real-world attackers participate is difficult.
>
> **[Regret Calculation]** Note that a testing scenario is difficult to design in our setting given the need for black-hat attackers who actually attack the system. The policy learned in training converges to the optimal movement policy and thus, during online deployment, we do not expect to have any regret. Further, from the figures shown, the agent yields the optimal reward after a few episodes. Hence, the area between the line averaging the optimal reward and the reward value curves for each agent is a measure of the regret during offline training.

---

### Official Review · AnonReviewer3 · 2020-10-28
**The problem is interesting, but the contribution is incremental. Borderline paper.**

**Rating:** 5
**Confidence:** 4

**Review:**

High-level summary:
This paper introduces a Bayesian Stackelberg Markov Game (BSMG) model that considers a defender’s uncertainty over attackers’ types when implementing defensive strategies. It also proposes to use a Bayesian Strong Stackelberg Q-learning method to learn defense policies by first simulating an adversary to obtain feedback of an attack and then computing the Bayesian Strong Stackelberg Equilibrium for the BSMG with a solver. In this way, this work relaxes the assumption that the defender knows attackers’ types in existing game-theoretic models for moving target defense.

Strength:
This paper proposes a game-theoretic model for MTD that learns adversary types via repeated interactions with a simulated attacker. It introduces a Bayesian Strongly Stackellberg Q-learning method that converges to the Bayesian Strong Stackelberg Equilibrium of the BSMG. Empirical results show that the proposed method has advantages over several baselines, such as static policies (URS) and adaptive policies (e.g., B- EXP-Q and B-Nash-Q).

Weakness:
The BSMG model itself is incremental since it does not provide any additional interesting insights other than adding the Bayesian and Stackelberg assumption into a Markov game. Regarding the solution, it is unclear to me why vanilla Q-learning instead of other advanced RL algorithms (say, the sample efficient RL variant STEVE by Beckman et al. 2018) is appropriate in their solution (especially when sample efficiency is important here). In the experiments, while it is helpful to compare BSS-Q with several existing baselines, it is unclear whether the performance of BSS-Q is comparable to the Bayesian Stackelberg Game model when the defender has complete information about the attackers’ types. This comparison will generate insights into to what extend the knowledge of attackers’ types influences the effectiveness and efficiency of the defender’s defense mechanisms.

Minor:
Figures 2 and 4 (also Figure 6  in the appendix) are barely legible. The fonts are too small.

---

> ### Author Response · Authors · 2020-11-22
> **Rebutting concerns about novelty and comparison of results to model-based inference methods**
>
> We thank the reviewer for their comments and are glad that they found our game-theoretic modeling, our proposed learning approach and its (theoretical and empirical) efficacy as key strengths of the paper. We now rebut the arguments made by them against the paper and clarify some of the points made in the paper.
>
> The rewards obtained in the context of Intrusion Detection System placement using Moving Target Defense are equal to the rewards obtained if one were to use a model-based strategy inference approach (eg. Bellman back-up with the notion of Strong Stackelberg Equilibrium). We do not plot it in Figure 4 for ease of readability. For the case of web-applications, BSMG provides a more general model, and thus, the strategy learned is better than the optimal stage-agnostic strategy shown in Figure 2. We will add some of these points and make the figures more legible in the revised version of the paper.
>
> **[Novelty]** The trend in the use of game-theoretic techniques for real-world security domains has relied on model-based inference approaches for finding movement strategies at equilibrium [1]. Our work is a first step to move to a model-free learning approach, an attempt to convince the stakeholders that these approaches work equally well in comparison to existing model-based inference mechanisms. Hence, we choose a Q-learning style approach for which we can prove convergence to a Strong Stackelberg Equilibrium. (Also, please see our response to Reviewer 4).
>
> [1] Sengupta, S., Chowdhary, A., Sabur, A., Alshamrani, A., Huang, D. and Kambhampati, S., 2020. A survey of moving target defenses for network security. IEEE Communications Surveys & Tutorials.

---

### Author Response · Authors · 2020-11-23
**Rebuttal Version Change Log**

We thank the reviewers for their questions, comments, and suggestions. Here is the list of changes made to the rebuttal version.
- [R1] Code uploaded.
- [R1] Caption changed for Figure 3.
- [R2] Discussion on in-state vs in-episode attacker type sampling.
- [R3] Figure 2, 4 and 6 made more legible.
- [R3] Comments on comparison of our model-free learning to existing model-based inference methods.
- [R4] Proof sketch in the main body of the paper.
- [R1,R4] Grammar and typographical error correction.

---

### Decision · Program_Chairs · 2021-01-07
**Final Decision**

**Decision:**

Reject

**Comment:**

The paper proposes a new game-theoretic model, Bayesian Stackelberg Markov Game (BSMG), for designing defense strategies while accounting for the defender's uncertainty over attackers' types. The paper also proposes a learning approach, Bayesian Strong Stackelberg Q-learning (BSS-Q), to learn the optimal policy for BSMGs. It is shown that BSS-Q converges to an equilibrium asymptotically. Experimental results are provided to demonstrate the effectiveness of BSS-Q in the context of web application security. Overall this is an interesting approach and an important direction of research. However, the reviewers raised several concerns, and there was a clear consensus that the paper is not yet ready for publication. The specific reasons for rejection include the following: (i) the experimental results are not presented with sufficient clarity, no statistical significance tests are performed, and the choice of baselines is weak; (ii) the contributions are not sufficiently broad, the learning process described in the paper is unclear, and the framework requires a strong assumption of knowing the attackers' distributions. I want to thank the authors for actively engaging with the reviewers during the discussion phase. The reviewers have provided detailed feedback in their reviews, and we hope that the authors can incorporate this feedback when preparing future revisions of the paper.